# Challenges and Opportunities for Post-COVID Pulmonary Disease: A Focused Review of Immunomodulation

**DOI:** 10.3390/ijms26083850

**Published:** 2025-04-18

**Authors:** Steffi Verbeeck Mendez, Isabella L. Do Orozco, Guadalupe E. Gavilanez-Chavez, Arnulfo Hernán Nava-Zavala, Maria G. Zavala-Cerna

**Affiliations:** 1Facultad de Medicina, Universidad Autónoma de Guadalajara, Guadalajara 45129, Mexico; steffi.verbeeck@edu.uag.mx; 2Koch Institute for Integrative Cancer Research, Massachusetts Institute of Technology, Cambridge, MA 02139, USA; iod@mit.edu; 3Hospital General Regional 46, Órgano de Operación Administrativa Desconcentrada Jalisco, Instituto Mexicano del Seguro Social, Guadalajara 44329, Mexico; dragavilanez@gmail.com; 4Unidad de Investigación Epidemiológica y en Servicios de Salud, Centro Médico Nacional de Occidente Órgano de Operación Administrativa Desconcentrada Jalisco, Instituto Mexicano del Seguro Social, Guadalajara 44329, Mexico; navazava@yahoo.com.mx; 5Programa Internacional de Medicina, Universidad Autónoma de Guadalajara, Guadalajara 45129, Mexico; 6Departamento de Inmunología y Reumatología, Hospital General de Occidente, Secretaría de Salud Jalisco, Zapopan 45170, Mexico

**Keywords:** post-COVID syndrome, coronavirus infection, post-COVID-19 pulmonary fibrosis, immunomodulation

## Abstract

The resolution of the recent COVID-19 pandemic still requires attention, since the consequences of having suffered the infection, even in mild cases, are associated with several acute and chronic pathological conditions referred to as post-COVID syndrome (PCS). PCS often manifests with pulmonary disease and, in up to 9% of cases, a more serious complication known as post-COVID-19 pulmonary fibrosis (PC19-PF), which has a similar clinical course as idiopathic pulmonary fibrosis (IPF). Generating knowledge to provide robust evidence about the clinical benefits of different therapeutic strategies to treat the pulmonary effects of PCS can provide new insights to amplify therapeutic options for these patients. We present evidence found after a scoping review, following extended PRIMSA guidelines, for the use of immunomodulators in pulmonary PCS. We start with a brief description of the immunomodulatory properties of the relevant drugs, their clinically proven efficacy for viral infections and chronic inflammatory conditions, and their use during the COVID-19 pandemic. We emphasize the need for well-designed clinical trials to improve our understanding the physiopathology of pulmonary PCS and PC19-PF and also to determine the efficacy and safety of candidate treatments.

## 1. Introduction

The coronavirus disease 2019 (COVID-19) pandemic, caused by infection with SARS-CoV-2, represented a significant global challenge [1]; even though mortality was effectively decreased (3–5%), additional challenges arose, since post-COVID Syndrome (PCS) emerged as a new health problem worldwide. Although many COVID-19 cases resolve without complications, it has been reported that over 70% of survivors will have multiple complications in one or more organs up to four months after the initial onset of symptoms [2].

PCS encompasses various manifestations related to cardio-pulmonary, gastrointestinal, dermatological, and neurological systems. Nevertheless, there is no consensus on a well-established definition; the World Health Organization (WHO) defines this condition as “The continuation or development of new symptoms 3 months after the initial SARS-CoV-2 infection, with these symptoms lasting for at least 2 months with no other explanation”, whereas the Center for Disease Control and Prevention (CDC) in the USA defines it as an umbrella term for the wide range of physical and mental health consequences that are present for four or more weeks after SARS-CoV-2 infection. Other terms used by academia include: post-COVID-19 condition, post-acute sequelae of SARS-CoV-2 infection (PASC), post-COVID syndrome, chronic COVID-19, and persistent post-COVID symptoms [3].

Irrespective of the term, more than 150 symptoms have been associated with PCS, most commonly dyspnea, persistent cough, chest discomfort, and exacerbations of previous pulmonary pathologies such as asthma and COPD. The cardiovascular symptoms include tachycardia and coagulopathies; other symptoms are sleepiness, fatigue, myalgias, paresthesia, loss or change in appetite, anxiety, and mood changes [4,5], and these symptoms can last for more than six months [4,6,7,8]. One of the more serious complications is post-COVID-19 pulmonary fibrosis (PC19-PF), which has a similar clinical course as idiopathic pulmonary fibrosis (IPF), characterized by worsening lung function and carbon monoxide diffusing capacity, and it can lead to early death because of increased hospitalization secondary to deteriorating lung function [9].

Around 10–20% of patients infected with SARS-CoV-2 develop PCS, and the CDC suggests that 7.5% of the adult population has PCS symptoms [10]. Patients with PCS have a negative RT-PCR test result, although some may have a positive fecal array. Patients with prolonged ICU stays and mechanical ventilation, who experience long-term consequences attributed to the respiratory support, should not be considered as having PCS [6,11,12].

Several attempts have been proposed to identify attributable risk factors for the development of PCS; however, the results are conflicting in terms of age and comorbidities, probably due to distinctive study designs [5,8,13]. Some common findings are female sex (3:1 ratio), as well as a high initial viral load or late viral clearance, having >5 symptoms during the acute-COVID stage or a severe initial disease including progression to ARDS, including the severity of hypoxia. However, in their study of a cohort of 929 patients with PCS, Patton et al. found an association with the male sex, preexisting heart failure, and hypertension [14].

The pathophysiology of symptom persistence in PCS is complex and partially described; however, there is evidence of several organs targeted by virus tropism, and particularly the affinity for angiotensin converting enzyme-2 (ACE-2) receptors, which have been shown to spread widely in the human body [3]. Important for the pathogenesis of PCS is the severity of the initial infection, since patients with severe COVID experience a systemic proinflammatory immune response with elevated levels of IL-1, IL-2, IL-6, and IL-7, tumor necrosis factor (TNF)-a, GM-CSF, macrophage inflammatory protein 1-a, C-reactive protein (CRP), ferritin, and D-dimer. This is known as a cytokine storm or cytokine release syndrome (CRS), and it is responsible for the immunopathology behind the development of severe complications including sepsis, shock, respiratory failure, acute respiratory distress syndrome (ARDS), multiorgan system dysfunction, and death [2,15,16]. Pulmonary damage, characteristic of COVID-19, involves the direct infection of type 2 pneumocytes by SARS-CoV-2, but this also suggests a macrophage activation syndrome (MAS)-like disease, with endothelial activation, which explains why severely affected patients experience a higher degree of pulmonary microvascular events [17]. The systemic inflammatory response syndrome could be the underlying cause of the development of an acute respiratory distress syndrome or tissue damage and organ dysfunction, which, if persistent, can activate catabolic pathways and PCS [18]. Persistent SARS-CoV-2 RNA in lung and gastrointestinal tissue can cause a chronic proinflammatory state, with the suppression of IL-10 and the activation of IL-1, IL-6 and TNF-a [19].

In general, treatment recommendations for COVID-19 are mostly centered on the use of potent anti-inflammatory drugs, such as dexamethasone for patients progressing to severe illness, which is associated with decreased mortality [20]. This confirms the benefit of immunomodulatory agents during the acute phase of COVID-19; however, the indication for this type of drugs in the treatment of pulmonary affection during PCS remains unknown [21]. Treatments for PCS so far involve only symptomatic therapy; there is a high potential for new drugs to be added with the purpose of reducing the inflammatory and immunological impact of these sequelae [20]. For this purpose, we underwent a scoping review, to evaluate and summarize current pulmonary PCS treatments and the potential use of immunomodulatory agents. To our knowledge, the use of non-typical immunomodulators, such as macrolides, for the treatment of pulmonary PCS has not been reviewed before.

## 2. Materials and Literature Search Results

A literature search evaluated the existing published information on macrolides’ use for COVID-19 treatment and viral respiratory infections. For this purpose, we used a combination of terms related to viral respiratory infections and macrolides.

Our search strategy was performed using Medline (via PubMed) to identify articles published in English or Spanish. Keywords were selected using MeSH on Medline, including the following: “COVID-19”, “respiratory viral infections”, “immunomodulation”, “cytokine storm”, “severe acute respiratory syndrome”, “SARS-CoV-2”, ”post-COVID Syndrome”, and “Pulmonary fibrosis”.

The following types of articles were included: observational, randomized clinical trials, and reviews. We excluded in vitro and experimental studies in animals and case reports and studies in humans with non-respiratory infections or non-viral respiratory infections. Data were imported to an EndNote X9 library, where duplicates could be identified and removed. Then, articles were screened carefully by two independent researchers following the extended PRISMA guidelines for scoping reviews [18].

In total, 60 articles were found after reviewing their relevance for the present review, and 52 articles were finally included in the final draft based on relevance. Additional articles were added from references of the selected articles.

## 3. Discussion

### 3.1. PCS-Related Pulmonary Sequelae

COVID-19 infection is characterized by altered epithelial cells, interstitial lung disease, and progression to fibrosis [22]. Acute SARS-CoV-2 has a different array of possible symptoms, ranging from asymptomatic to systemic hyperinflammation [23]. The pulmonary long-term sequelae caused by COVID-19 are the consequence of direct injury to the alveoli, endothelium or pulmonary vasculature, with diffuse alveolar damage being the precursor to fibrotic changes and a reduction in the lungs’ capacity for diffusing carbon monoxide (DLCO) [24]. Previously attributable risk factors for the development of PC19-PF include lower BMI (*p* = 0.021), longer hospital stay (mean: 28, *n* = 80, *p* = <0.0001), ICU care during hospitalization (*p* < 0.0001), and a higher level of inflammatory markers (CRP, *p* = 0.059) [25].

Before the COVID-19 pandemic, pulmonary fibrosis was the most severe sequelae of interstitial lung disease, and it most frequently developed after epithelial and diffuse alveolar damage in patients with previously known connective tissue disorders, chronic granulomatous disease, and/or respiratory infections [26]. The resolution of alveolar and epithelial damage among patients with COVID-19 is not uniform and depends largely on the acute-phase characteristics of the lung disease (the organizing pneumonia pattern versus fibrinous organizing pneumonia) and whether the patient had a previously diagnosed lung condition [24].

Furthermore, not all individuals who contract COVID-19 will develop fibrosis. It has been estimated that only 9% of patients with severe or critical disease will develop fibrotic-like changes [27], and, among those, up to 42% will persist with manifestations and clinical signs of pulmonary fibrosis two years after the acute infection [28].

Due to its impact on quality of life, the most important of post-COVID-19’s long-term sequelae is PC19-PF, which exhibits fibroblast persistence and the massive deposition of collagen, elastin, and other extracellular matrix components. Under normal circumstances, these are removed, as well as the pulmonary edema fluid; however, in the context of PCS, persistent immune activation, either secondary to absence in viral clearance or damage reluctance, will drive uncontrolled fibroproliferation with the upregulation of the profibrotic pathways and downregulation of the antifibrotic pathways. The former has been confirmed after finding an upregulation in the expression of TGF-b in the lungs of patients following SARS-CoV-2 infection [29]. During acute lung injury (ALI)/ARDS and continued viral infection, there is a redox imbalance due to reduced antioxidant defense and increased reactive oxygen species (ROS) production, which causes oxidative damage and stimulates the differentiation of alveolar epithelial cells to fibroblasts, with increased extracellular matrix deposition [30].

Elastin fibers are found abundantly in the pulmonary epithelium, and they are essential for the expansive and compliance properties of the alveoli. Elastin fibers are a substrate for neutrophil elastase, and, when activated, they release a considerable amount of elastin degrative products (EDP), which induce tissue and interstitial inflammation, potentially leading to fibrosis. This pathophysiological process may coexist with injury-associated fibrosis, idiopathic pulmonary fibrosis (IPF), and chronic obstructive pulmonary disease (COPD). This elastolytic activity has been shown to play an important role in PCS and PC19-PF [31,32]. Gerayeli et al. confirmed the important role of neutrophils in PC19-PF, finding a unique set of these cells related to degranulation and to the so called “cytokine storm” [33].

PC19-PF diagnosis requires that one of the following is present at the three-month follow up: (1) persistent COVID-19 respiratory symptoms; (2) hypoxemia at rest and/or during exercise not attributable to any other cause; (3) restrictive ventilatory defect (forced vital capacity [FVC) of <80% of the predicted normal and/or DLCO of <80% of the predicted normal); and (4) pulmonary imaging changes such as ground glass opacities, subpleural and/or fibrotic bands, and consolidations, as well as fibrotic-like changes on a follow-up CT of the thorax [34].

Because of high ECM accumulation, studies have attempted to find serum biomarkers of pulmonary fibrosis, such as laminin (LN), type IV collagen (IV), type III procollagen N-terminal peptide (PIIINP), and hyaluronic acid (HA), which display increased levels compared to healthy individuals, and, in the case of hyaluronic acid and PIIINP, are correlated with a worse disease course [35].

The viral persistence of SARS-CoV-2 may play an important role in developing PCS, but few studies have associated the initial viral load present in the acute phase with long-term outcomes. Tromp et al. studied a cohort of 4054 patients and found that higher viral loads at admission were associated with an increased risk of death and rehospitalization after discharge. Other studies correlated higher SARS-CoV-2 initial viral loads with in-hospital mortality or intubation [36]. The reactivation of chronic viral infections has also been investigated by Maguire et al., who found specifically that *Herpesviridae* and *Anelloviridae* reactivate in the setting of acute severe COVID-19 infections; this finding was based on a prospective study of >1000 hospitalized COVID-19 patients. Additionally, they found that the use of azithromycin (AZM) was associated with a decrease in cytomegalovirus (CMV) in the nasal compartment, which can lead to an increase in T-cell activation. In contrast, Epstein–Barr Virus (EBV) can lead to an increase in IL-6 and CXCL10, which are known to be correlated with COVID-19 severity. Viral replication may also contribute to oxidative stress and cellular damage, particularly EBV reactivation, which is more pronounced in severe COVID-19 infections, and this has been positively correlated with a higher incidence of PCS [37].

These pulmonary changes can be identified using CT scans, during follow-up or after persistent symptoms, most of which manifest with restrictive lung function characteristics. On CT scans or high-resolution CT (HRCT), the most common characteristic PC19-PF findings can initially be identified as bilateral ground glass opacities, traction bronchiectasis, reticulation, and the presence of fibrotic bands in the parenchyma [22]. The ground glass opacities, also called ‘snowstorm’, may disappear with time or consolidate with the development of fibrosis, which is known as the ‘honeycomb’ image and is characteristic of PCS19-PF. Additionally, air trapping on expiratory CTs can be observed (Figure 1). It is important to emphasize that no group of findings is typical, meaning that there can be different findings that vary from patient to patient [38].

Babar et al. found severe abnormalities in CT scans up to two years after the acute-phase COVID-19. These include fibrotic-like changes, bronchiectasis, and reticulation, which show no significant changes over time (24-month follow up) and have higher incidence in patients with severe COVID-19 infections: 75% of CT abnormalities occur in severe cases versus 38%, as compared to a study conducted by Watanabe et al. with a 12 month follow-up. Although there is the presence of fibrosis, not all changes indicate progressive pulmonary fibrosis, which is why CT should be complemented with PFTs [39,40]. Han et al. reported 62% of CT anomalies at six months, with a 35% prevalence of fibrotic changes, versus 72% according to Caruso et al. These interstudy discrepancies highlight the fact that research is still ongoing and there is still a vacuum in the current knowledge about PCS [41,42]. A prospective cohort study in 237 COVID patients, with a follow up of 18–24 months, reported that 80 (35%) of patients persisted with fibrotic-like lesions at the end of the follow-up period; these patients had an increase in architectural distortion from 21% at baseline to 28% at the end of follow up, as well as traction bronchiectasis (25% vs. 34%), with up to 44% of patients having a restrictive pattern with persistent functional impairment and reduced diffusion capacity [25]. These pathological fibrotic lesions have also been identified in similar viral respiratory diseases such as avian influenza, 2003-SARS-CoV, and Middle East respiratory syndrome (MERS), and the data suggest that patients who have recovered from 2003-SARS and MERS have a decreased DLCO, with a 20-year gap between initial infection and last spirometry [43,44]. Taken together, the mechanisms for lung fibrosis in PCS may be similar to those of IPF; however, a detailed analysis of the HRCT, revealing a honeycombing pattern in a limited proportion of patients, might suggest a different clinical scenario, which highlights the importance of a better targeted treatment.

### 3.2. Current PCS Therapy

Treatment for PCS still lacks proper standardization and is exclusively symptomatic; little effort has been made to alleviate other associated pathologies and the root of the disease in order to reduce morbidity and return individuals to their pre-COVID state [45]. Some treatments include mast-cell stabilizers to control their access to inflamed tissue and consequent cytokine production (IL-1, IL-6 and TNF-a) [46]; moreover, it has been suggested that treatment for idiopathic pulmonary fibrosis could benefit patients with PC19-PF [47].

Antifibrotic medication inhibits tyrosine kinase, leading to the suppression of fibroblasts and myofibroblasts; these include pirfenidone and nintedanib, which have shown both radiological and clinical improvements, although reported use has only been described in selected case reports, and no large trials have been conducted [47]. Pirfenidone is an oral antifibrotic indicated for IPF treatment, lengthening progression-free survival and delaying the decline in pulmonary function. In COVID-19, pirfenidone downregulates the expression of TFG-B, thereby reducing the release of extracellular proteins and fibroblast activity and conversion [9]. In this setting, it has been proven that, when combined with standard treatment versus standard care, a higher percentage of patients in the pirfenidone group did not develop pulmonary fibrosis (21.3% in the standard care group versus 5.7% in the pirfenidone group) [48]. Additionally, Tanvir et al. performed an open-label pilot trial with 36 patients with severe COVID-19. They were divided into two groups: the first one received pirfenidone (*n* = 17) and the second received corticosteroids (CS) (*n* = 19). They found that pirfenidone has a better outcome in terms of preventing pulmonary fibrosis in comparison to CS [49]. Although these types of therapies sound promising, additional studies must be performed to evaluate their long-term benefits and increase the number of patients for validity [50].

Anti-inflammatory drugs and systemic CS have become common treatments for those patients who continue to exhibit hypoxia; however, certain restrictions are associated with the time of use and side effects when the treatment is either given orally [51] or inhaled [52]. Systemic CSs are the most commonly used therapy in case of persistent interstitial lung involvement, leading to the resolution of inflammation and fibrosis by downregulating the various cytokines and molecules involved. Nonetheless, it has been proven that corticoid therapy is the most effective when treating patients before they manifest fibrotic changes. However, in the studies conducted by Mizera et al. and Dhooria et al., patients prescribed CS experienced significant radiological improvement, without considering pulmonary function tests [53]. Compared to antivirals prescribed during the acute infection, CS did not protect against the development of PCS [54]. The suggested indication criteria for CS treatment rely on previous studies with patients experiencing significant parenchymal involvement, persistent subjective symptoms, resting desaturation or the same during a six-minute walk test. Dosage should be individualized, with a starting dose of 0.5 mg/kg of prednisolone in patients at risk of adverse events [53]. N-acetylcysteine, commonly used for acetaminophen overdose, has also proven to be valuable for the treatment of PC19-PF. It replenishes glutathione levels, which, as mentioned above, have been found to be diminished in COVID-19 and cause redox imbalances that can lead to pulmonary fibrosis [55,56].

The only definitive treatment for end-stage PF is a lung transplant; nevertheless, it is associated with high mortality, morbidity, and poor long-term survival, and to date there are no long-term reports available about follow-up of PCS transplanted patients [57]. Patients who have undergone lung transplants with acute COVID-19-induced ARDS and severe lung damage showed no improvement despite prolonged mechanical ventilation and extracorporeal membrane oxygenation. Bharat et al. reported a positive outcome in a cohort of 12 transplanted patients, with good short-term survival; interestingly, the pathology reports of the damaged lungs showed early signs of pulmonary fibrosis in these patients [58].

Gazzaniga et al. explored the use of allogenic mesenchymal stromal cells (MSCs) as a therapeutic approach. Following a compassionate release, they treated six adults on mechanical ventilation for moderate COVID-19-related ARDS with IV MSCs. They showed that patients with moderate ARDS could benefit from this therapy with an increase in the PaO_2_/FiO_2_ ratio > 200 after the first or second infusion; nonetheless, there was no benefit in patients with severe ARDS and an increased immune response was noted [59]. Natural substances such as flavanone derivatives have also been investigated, as it has been seen that they have immunomodulatory effects; in the COVID-19 setting, they can reverse the cytokine storm by inhibiting IL-6. Liu et al. suggested that naringin and naringenin are inhibitors of PC19-PF, comparing their effectiveness to that of monoclonal antibodies. Fibrosis is prevented by inhibiting TFG-B expression and regulating collagen formation [60].

### 3.3. PCS and Immunomodulatory Treatment with Macrolides

Macrolides are a group of antibiotics the activity of which is related to the macrolide ring and leukocyte accumulation; their mechanism of action is the inhibition of protein synthesis after receptor binding on the bacterial 50S ribosomal subunit. This group of antibiotics includes erythromycin (ERY), azithromycin (AZM), and clarithromycin (CAM). Apart from its bacteriostatic/bacteriocidic effects, this pharmacological group has certain widely known immunomodulating and anti-inflammatory effects related to several cells and pathways in chronic inflammatory pulmonary diseases with pulmonary fibrosis, such as COPD, cystic fibrosis (CF), and asthma. Their immunomodulation has been widely studied and was first described in the 1950s. In the 1970s, it was found that patients with diffuse pan-bronchiolitis treated with low-dose ERY over a prolonged time exhibited clinically beneficial effects (reduction of exacerbation rate by 30%) and significantly improved survival [61].

Macrolides’ anti-inflammatory effect is modest compared to other drugs, generating questions about their real-world use. However, their immunomodulatory effects are worthy of consideration for persistent chronic inflammatory conditions associated with viral infections [62]. Some previously described mechanisms include the downregulation of cytokines, including IL-1, IL-6, and IL-8, which directly influence phagocyte activity by modifying chemotaxis, phagocytosis, oxidative burst, and the production of type I interferons [63]. Additionally, CAM suppresses the production of matrix metalloproteinase-9 and decreases pathological pulmonary changes [64]. CAM additionally had a boost for mucosal immunity [65]. Their role in impeding inflammatory cytokines synthesis is due to their interaction with intracellular mitogen-activated protein kinase (MAPK) and the NF-kB transcription factor [65] (Figure 2). CAM is indicated for managing chronic inflammatory lung diseases, such as bronchiectasis, chronic obstructive pulmonary disease, chronic rhinosinusitis, diffuse bronchiolitis [66], and children with asthma [67]. A prophylactic low dose has also been indicated for patients with bronchiectasis (not associated with cystic fibrosis) due to the reduction in Th17 cells and IL-17 in peripheral blood, as well as improvements in PaCo2 and FEV1, demonstrating a possible role in the regulatory balance of helper T cells [68].

AZM’s postulated antiviral properties could be synergistic with antiviral treatment; such observations were derived from experimental studies with Zika virus [69], Ebola virus [70], and rhinovirus [71]. The unique immunomodulatory pathway on which AZM acts involves downregulating proinflammatory pathways via NF-kB-like corticosteroids. However, a significant difference between these two is that AZM does not completely lead to immunosuppression but enhances the functional aspects of regulation and repair. They also directly affect neutrophil function, which plays a destructive role in pulmonary diseases, functioning as an anti-fibrotic and anti-inflammatory agent and decreasing the neutrophil oxidative burst [72]. During the COVID-19 pandemic, multiple studies evaluated macrolide use in acute COVID-19; nonetheless, various trials found little to no effectiveness in their use or were terminated prematurely.

Despite these results, the emerging data suggest that active virus replication and viral persistence in the upper respiratory tract are associated with the severity of the disease [73,74]. Only a few clinical trials included in their outcomes a decrease in the viral load, especially when CAM was administered in the first five days of symptom onset, which may be the result of the immunomodulatory effects of the antibiotic, towards the Th1 type of response required for antiviral immunity [75]. This enables the consideration of benefits with use of CAM early in the disease, for a limited time, and for the single purpose of decreasing viral replication and inflammation, while boosting mucosal immunity (Figure 2). Furthermore, the development of PCS in patients who received macrolides as part of their initial treatment for acute COVID-19 could be investigated to analyze this possible preventive effect for PCS, although this remains to be proven.

### 3.4. Future Perspectives

Although these theories remain mostly unproven for the development of PCS, the description includes three scenarios: altered anticoagulation, immune dysregulation, and viral persistence and/or reactivation [7,76]. Some important work has been conducted to increase our understanding of the pathology. The RECOVER (Research COVID To Enhance Recovery) initiative aims to fund multicenter research to discover why some individuals have persistent symptoms of COVID-19, independent of whether they experienced mild–moderate or severe COVID-19 [77].

The controversies over the use of macrolides for acute COVID-19 have been resolved; however, a large knowledge gap remains regarding their use for pulmonary PCS. According to the limited information available with respect to PCS pathophysiology, patients could benefit from the administration of immunomodulatory drugs that can limit viral replication and the chronic inflammatory response. The immunomodulatory functions of macrolides that make them candidates for PCS prevention or treatment during acute infection include their anti-inflammatory and antiviral properties; additionally, by boosting the mucosal immune response, they exhibit characteristics that could prevent the development of PCS (Figure 1). However, in the case of pulmonary PCS, additional effects include the promotion of epithelial integrity and the prevention of pulmonary fibrosis by decreasing the proliferation of fibroblasts, reducing the production of collagen and the expression of TGF-b [78]. Additionally, CAM has shown to be a highly effective mast cell stabilizer [79]. Furthermore, CAM has effectively decreased exacerbations in several lung conditions with chronic inflammation [67,80], as well as improving viral infections [63,81,82].

Antivirals have become drugs of interest in relation to PCS treatment, due to the evidence suggesting the persistent presence of SARS-CoV-2 in anatomical reservoirs months after the infection. The recover vital trial is a prospective, multicenter, multiarm, double-blind, randomized, and controlled platform trial, which researches the effects of nirmatrelvir and ritonavir to treat patients with long COVID. However, ongoing active viral replication has yet to be confirmed, meaning that antivirals’ use in the treatment of PCS might be limited.

Several questions remain unanswered that might improve our understanding and help in the design of targeted therapy. If the virus-associated material persists, could this be a factor that triggers the activation of the immune system in PCS? Does this chronic activation lead to immunosenescence in patients with PCS? Can we improve the treatment of PCS by promoting immunomodulation with macrolides? These questions should be the focus of new research to find a more suitable therapy for PCS.

## 4. Conclusions

The post-pandemic disease burden has been described as a post-COVID syndrome and includes a variety of symptoms, including post-COVID-19 pulmonary disease and pulmonary fibrosis (PC-19PF), with the last being less prevalent but more serious. Preventing and treating this syndrome has been challenging, as we lack a full explanation for the exact pathophysiology.

There is an urgent need to implement well-designed clinical trials to improve our understanding of the efficacy and safety of additional candidate treatments. Immunomodulatory drugs can be used to treat post-COVID pulmonary disease and prevent PC-19F, as their effects can be beneficial in terms of disease progression and management. Nonetheless, their use is limited due to the lack of investigation of the correlation of viral load with acute-phase treatment, as well as their effects in preventing pulmonary fibrosis. Their use could decrease the pathological consequences of an inflammatory response, favoring a better innate immune response and cellular chemotaxis, as well as boosting mucosal immunity, which has been observed in other respiratory viral infections when combined with antiviral treatment. However, caution is advised due to evidence of regression in fibrotic changes with a non-interventional approach in some cases.

Future clinical studies with randomization and follow-up are needed to correlate the initial disease burden, viral load, and acute-phase treatment with the post-COVID sequelae and pulmonary fibrosis, as their long-term effects on pulmonary PCS remain elusive.

## Figures and Tables

**Figure 1 ijms-26-03850-f001:**
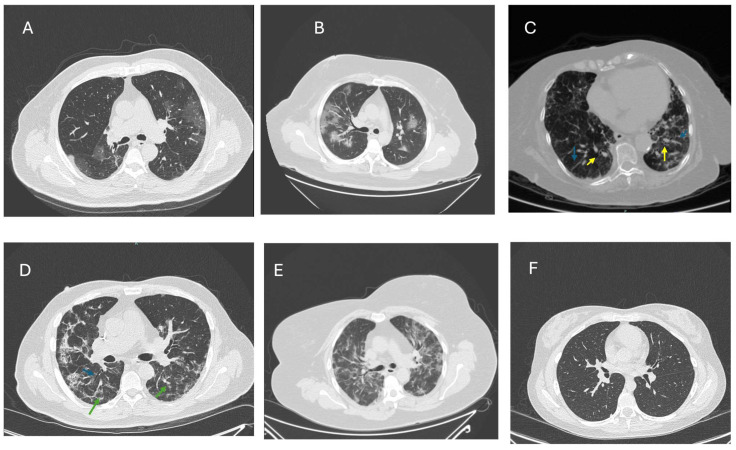
Common findings on CT scans from patients with COVID-19 and PC19-PF. (**A**) CT scan from a patient with early pulmonary changes associated with acute COVID with bilateral, multifocal peripheral cotton-like lesions. (**B**) Patient with acute COVID-19: the CT scan shows widespread, bilateral ground glass opacities (GGO) in combination with thickened interlobular and intralobular lines, commonly referred as crazy paving. (**C**) Patient with PCS (six months after the acute infection) with dilated vessels on affected areas (yellow arrows) and septal thickening with a characteristic honeycomb appearance; band atelectasis (blue arrows) is also observed in this image. (**D**–**F**) belong to the same patient with PCS19-PF. (**D**) Patient with PCS19-PF with fibrotic bands (green arrows) and the characteristic honeycomb, with traction bronchiectasis (blue arrow). (**E**) Three months later, the patient developed a viral influenza infection, and the CT scan shows a combination of fibrosis persistence with multiple focal areas of consolidation, representative of an active infection. (**F**) Six months after the infection with influenza, an almost complete resolution of the fibrotic changes is observed, with only a few areas of incipient fibrosis.

**Figure 2 ijms-26-03850-f002:**
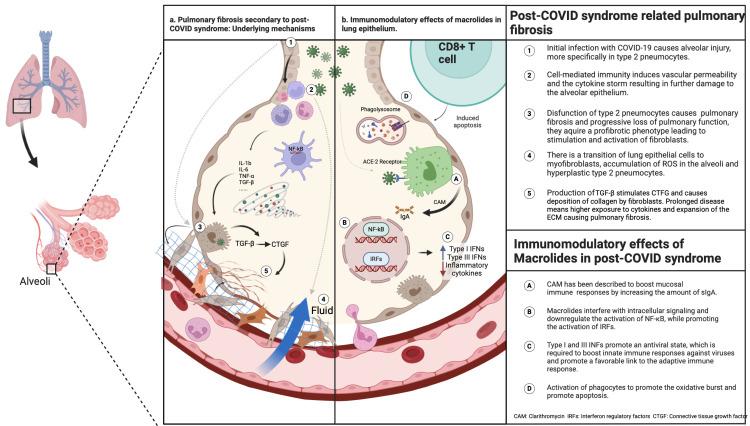
PCS-related pulmonary fibrosis and potential immunomodulatory effects of macrolides [32]. Created in BioRender. Zavala, M. (2025) https://BioRender.com/s10u769.

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
