# Peer review of "Challenges and Opportunities for Post-COVID Pulmonary Disease: A Focused Review of Immunomodulation"

_ijms, 2025, doi:10.3390/ijms26083850_

Round 1
Reviewer 1 Report
Comments and Suggestions for Authors
The review concerns an important issue of post - Covid pulmonary fibrosis. Immunological mechanisms responsible for non-resolving lung disease and the possibility of its treatment with immunomodulatory and/or antifibrotic drugs are discussed.
My critical remarks concerning the paper are as following:
- The authors state that post-Covid pulmonary fibrosis is a frequent problem. In my opinion, such conclusions are based on several publications that describe ANY post-Covid lung pathology, not specifying the prevalence of fibrotic lung lesions. Therefore, it is important to distinguish in the review, the HRCT features of fibrosis in post-Covid lung, such as reticular opacities, bronchial distortion and honeycombing, from the features of Covid-related pneumonitis, such as ground glass opacities and lung consolidations. The features of lung fibrosis are irreversible, on the contrary – features of post-Covid pneumonitis resolve (often spontaneously) within 1-2 years of observation.
The prevalence of ANY lung pathology in HRCT, 3 months from COVID diagnosis, is about 60%. But signs of lung fibrosis at that time frame are described in 4% of hospitalized patients, and possibly reach 8-9% of patients who survived severe/critical Covid pneumonia (J Clin Med 2025, 14, 347; Eur J Med Res 2024,29,585; Diagnostics 2024, 14, 2811).
- I would expect that the authors include in the review HRCT scans representing post-Covid lung pathology and specify the fibrotic opacities.
- Although the mechanisms of lung fibrosis in post-Covid patients may be similar to those observed in IPF, the clinical picture is different as honeycombing is observed in majority of IPF patients, and only in 3.8% of patients with post-Covid lung involvement .
- The above mentioned remarks influence the possible therapy proposed. The literature search should answer the questions: who is the candidate for treatment, what is the supposed treatment effect and what is the timing of drug use. In my opinion, therapy should concern the patients with the features of protracted post-Covid pneumonitis, especially if they present reduced exercise capacity as well as the decreased FVC and DLCO% predicted. The conclusions concerning the efficacy of such treatment should be cautious, as the spontaneous regression of the disease is also observed (J Clin Med 2025, 14,347).
Author Response
We would like to thank reviewers for their time, as their constructive criticism has improved the quality of our manuscript.
1. The authors state that post-Covid pulmonary fibrosis is a frequent problem. In my opinion, such conclusions are based on several publications that describe ANY post-Covid lung pathology, not specifying the prevalence of fibrotic lung lesions. Therefore, it is important to distinguish in the review, the HRCT features of fibrosis in post-Covid lung, such as reticular opacities, bronchial distortion and honeycombing, from the features of Covid-related pneumonitis, such as ground glass opacities and lung consolidations. The features of lung fibrosis are irreversible, on the contrary – features of post-Covid pneumonitis resolve (often spontaneously) within 1-2 years of observation.
The prevalence of ANY lung pathology in HRCT, 3 months from COVID diagnosis, is about 60%. But signs of lung fibrosis at that time frame are described in 4% of hospitalized patients, and possibly reach 8-9% of patients who survived severe/critical Covid pneumonia (J Clin Med 2025, 14, 347; Eur J Med Res 2024,29,585; Diagnostics 2024, 14, 2811).
RESPONSE: We stated that pulmonary fibrosis is an important complication, not because of its frequency, but rather to the impact of the condition in human health. To avoid misinterpretations, we have added 3 paragraphs with 8 new references, stating different sequelae and the information available to PCS related pulmonary fibrosis, including data on frequency and temporality.
2. I would expect that the authors include in the review HRCT scans representing post-Covid lung pathology and specify the fibrotic opacities.
RESPONSE: We have added information about CT scans on lines 210-219 as well as images from patients with COVID-19 and post-COVID associated pulmonary fibrosis.
3. Although the mechanisms of lung fibrosis in post-Covid patients may be similar to those observed in IPF, the clinical picture is different as honeycombing is observed in majority of IPF patients, and only in 3.8% of patients with post-Covid lung involvement .
RESPONSE: We have added information about frequency of PF in COVID patients and information about HRCT according to previous comments. We also added a sentence at the end of section 3.2 to summarize this clinical picture.
4. The above mentioned remarks influence the possible therapy proposed. The literature search should answer the questions: who is the candidate for treatment, what is the supposed treatment effect and what is the timing of drug use. In my opinion, therapy should concern the patients with the features of protracted post-Covid pneumonitis, especially if they present reduced exercise capacity as well as the decreased FVC and DLCO% predicted. The conclusions concerning the efficacy of such treatment should be cautious, as the spontaneous regression of the disease is also observed (J Clin Med 2025, 14,347).
RESPONSE: We have added information about candidates for treatment and timing of drug use. We also rephrased the conclusion taking into consideration spontaneous regression of the disease.
Reviewer 2 Report
Comments and Suggestions for Authors
This manuscript reviews the latest studies on the mechanism, epidemiology, and treatment of post-COVID pulmonary fibrosis. It is an interesting topic, and the manuscript is clear and well-structured. However, the following concerns should be addressed.
- The most significant issue is the novelty of the manuscript, as there are already reviews discussing post-COVID pulmonary fibrosis. The authors should clearly highlight how their review differs from existing literature.
- The manuscript would benefit from including data on the evaluation of disease severity and prognosis, particularly regarding the proportion of patients in whom this condition is reversible and those who experience relative long-term effects, as this is of significant clinical interest.
- Regarding treatment with macrolides, data on their use in post-COVID syndrome-PF should be included, as this section currently lacks a clear connection to the overall theme of the manuscript.
Author Response
We would like to thank reviewers for their time, as their constructive criticism has improved the quality of our manuscript.- The most significant issue is the novelty of the manuscript, as there are already reviews discussing post-COVID pulmonary fibrosis. The authors should clearly highlight how their review differs from existing literature.
RESPONSE: Thank you for your comment. We have added a sentence at the end of the objective to highlight the novelty of our paper.
2. The manuscript would benefit from including data on the evaluation of disease severity and prognosis, particularly regarding the proportion of patients in whom this condition is reversible and those who experience relative long-term effects, as this is of significant clinical interest.
RESPONSE: We added information related to the number of patients with long-term effects at the beginning of section 3.2.
3. Regarding treatment with macrolides, data on their use in post-COVID syndrome-PF should be included, as this section currently lacks a clear connection to the overall theme of the manuscript.
RESPONSE: We have added this information to the novelty of the paper, as well as information related to their use (mostly to prevent PF).
Round 2
Reviewer 1 Report
Comments and Suggestions for Authors
Dear Authors,
Thank you very much for the revised version of the manuscript. Nevertheless, I have still some remarks to the content. The included chest CT scans present typical evolution of lung opacities in COVID-related pneumonia. Figures D-F, belonging to the same patient document nearly complete regression of lung opacities. Therefore, I have doubts concerning the nature of these opacities, they were probably consolidated post-inflammatory lesions, that regressed during observation. Was it spontaneous regression, or the result of the therapy?
I agree that some of the lung opacities observed in post-covid patients may require therapy, but only the minority of them is confirmed as lung fibrosis. Maybe it is reasonable to modify the title? I propose “Challenges and opportunities for post-COVID pulmonary disease: a focused review on immunomodulation”. If you modify the title, do NOT forget to make similar corrections in the whole text.
The other remark concerns the abstract, please modify the fragments concerning the frequency of post-Covid pulmonary fibrosis as well as its comparison to IPF, it should correspond to the data included in the text.
Author Response
Thank you very much for the revised version of the manuscript. Nevertheless, I have still some remarks to the content. The included chest CT scans present typical evolution of lung opacities in COVID-related pneumonia. Figures D-F, belonging to the same patient document nearly complete regression of lung opacities. Therefore, I have doubts concerning the nature of these opacities, they were probably consolidated post-inflammatory lesions, that regressed during observation. Was it spontaneous regression, or the result of the therapy?
RESPONSE: Thank you for your comments. You raised an important question that we are not able to respond since we only had access to the imagen files and not the case itself. However, we do know that the opacities observed in the first figure D are a consequence of pulmonary post-COVID, in letter E the patient was diagnosed with an aggregated influenza infection, and letter F shows the resolution of the acute infection but also other previously observed fibrotic lesions. The reason for this, might be the result of therapy for the influenza infection and maybe reactivation of the immune response due to a real threat which was the influenza viral infection.
I agree that some of the lung opacities observed in post-covid patients may require therapy, but only the minority of them is confirmed as lung fibrosis. Maybe it is reasonable to modify the title? I propose “Challenges and opportunities for post-COVID pulmonary disease: a focused review on immunomodulation”. If you modify the title, do NOT forget to make similar corrections in the whole text.
RESPONSE: We agree with this comment, we changed the title, the information in the abstract and the body of the manuscript to refer to both pulmonary disease and pulmonary fibrosis, or in particular sessions to either pulmonary disease or pulmonary fibrosis.
The other remark concerns the abstract, please modify the fragments concerning the frequency of post-Covid pulmonary fibrosis as well as its comparison to IPF, it should correspond to the data included in the text.
RESPONSE: We included information in the abstract about the frequency of post-Covid pulmonary fibrosis according to the data included in the text.
Reviewer 2 Report
Comments and Suggestions for Authors
The revised manuscript is much better. I have no other comments.
Round 3
Reviewer 1 Report
Comments and Suggestions for Authors
Thank you for replying to all my remarks and for performing the revision accordingly